# Signatures of sliding Wigner crystals in bilayer graphene at zero and finite magnetic fields

Anna M. Seiler [1] ✉, Martin Statz [1], Christian Eckel [1], Isabell Weimer[1], Jonas Pöhls [1], Kenji Watanabe [2], Takashi Taniguchi [3], Fan Zhang [4] & R. Thomas Weitz [1] ✉

AB-stacked bilayer graphene has emerged as a fascinating yet simple platform for exploring macroscopic quantum phenomena of correlated electrons. Under large electric displacement fields and near low-density van-Hove singularities, it exhibits a phase with features consistent with Wigner crystallization, including negative dR/dT and nonlinear bias behavior. However, direct evidence for the emergence of an electron crystal at zero magnetic field remains elusive. Here, we explore low-frequency noise consistent with depinning and sliding of a Wigner crystal or solid. At large magnetic fields, we observe enhanced noise at low bias current and a frequency-dependent response characteristic of depinning and sliding, consistent with earlier scanning tunnelling microscopy studies confirming Wigner crystallization in the fractional quantum Hall regime. At zero magnetic field, we detect pronounced AC noise whose peak frequency increases linearly with applied DC current—indicative of collective electron motion. These transport signatures pave the way toward confirming an anomalous Hall crystal.

A recurring theme in condensed matter physics is the exploration of the ground states of a strongly correlated electron system under different conditions[1], particularly when a new system is established in experiment. One candidate state in two dimensions is the so-called Wigner crystal (WC) that minimizes the Coulomb interaction energy of itinerant electrons in the low-density limit[2]. In a WC, due to the dominance of Coulomb repulsion, the itinerant electrons arrange themselves below a melting temperature into a periodic lattice independent of the underlying atomic structure[2,3]. A central question in the field is how to identify the presence of Wigner crystallization, which is especially critical in situations where direct imaging of an emergent electron crystal is not straightforward[4,5]. An elegant method in this respect is electronic noise measurements: When a sufficiently high external in-plane electric field is applied across the sample, the electron lattice pinned by charge impurities can become mobile. This depinning triggers a sliding behavior, where the Wigner crystal phase experiences gradual, collective motion in response to the external perturbation[6–8]. This motion gives rise to an oscillatory current determined by the sliding velocity and the lattice constant or average domain size of the emergent WC, and this unique behavior is often detectable as noise in DC transport measurements[6–8]. Therefore, detecting an AC current with a frequency that is proportional to the applied bias current serves as a clear signature of sliding WC's[9–16].

To date, the investigation of noise in charge ordered states has mainly focused on quasi-one or two dimensional bulk materials, such as WC phases at large magnetic fields[6,12,15,17–21] and incommensurate CDW phases in transition metal dichalcogenides[9,10,22]. However, recent advancements have expanded the scope of exploring correlated electronic ordering in even simpler systems that neither require a magnetic field to quench the kinetic energy nor a moiré superlattice to

[1]1st Physical Institute, Faculty of Physics, University of Göttingen, Göttingen, Germany. [2]Research Center for Functional Materials, National Institute for Materials Science, Tsukuba, Japan. [3]International Center for Materials Nanoarchitectonics, National Institute for Materials Science, Tsukuba, Japan. [4]Department of Physics, University of Texas at Dallas, Richardson, TX, USA. ✉e-mail: seileran@phys.ethz.ch; thomas.weitz@uni-goettingen.de

trap the emergent electron crystal. Specifically, the naturally occurring rhombohedral-stacked multilayer graphene systems have emerged as an outstanding platform for studying strongly correlated electrons[5,23–30]. Not only is this evident when a large magnetic field is applied[5], but this has also been identified even at zero magnetic field when the low charge carrier density is tuned to the vicinity of van Hove singularities[23–25]. While magnetic field-induced WC's have been observed using high-resolution scanning tunneling microscopy (STM) in AB-stacked bilayer graphene (BLG)[5], compelling evidence for Wigner crystal states at zero magnetic field remains elusive in graphene systems, although bias spectroscopy and temperature dependent resistance measurements have provided first hints towards correlated phases consistent with such ordering in AB-stacked BLG[23,24] and rhombohedral pentalayer graphene[25]. To provide further evidence for Wigner crystallization in BLG with transport measurements, one smoking-gun experiment is to investigate the low-frequency noise generated by depinning and sliding of emergent electron lattices at zero magnetic field near the isospin Stoner phase regime[23,24,31] and at large magnetic fields near the fractional quantum Hall regime[5].

In this work, we perform low-frequency noise measurements on BLG to search for transport signatures of Wigner crystallization under both zero and high magnetic fields. At high magnetic fields, we observe enhanced low-frequency noise at small bias currents, with a frequency-dependent response characteristic of depinning and sliding of an electron lattice—consistent with prior scanning tunneling microscopy observations of Wigner crystallization in the fractional quantum Hall regime. At zero magnetic field, we detect pronounced AC noise whose peak frequency scales linearly with the applied DC current, indicating

collective electron motion, consistent with the presence of Wigner crystallization in AB-stacked bilayer graphene.

## Results

The BLG flake investigated in this study is encapsulated in hexagonal boron nitride (hBN) and equipped with graphite top and bottom gates and two-terminal graphite contacts (the device and its fabrication were detailed in ref. 23). This configuration enables to continuously tune the vertical electric displacement field $D$ and the charge carrier density $n$ (see Methods). All measurements, unless stated otherwise, were conducted in a dilution refrigerator at a base temperature of 10 mK using a standard lock-in technique (see Methods).

To establish the measurement technique in our BLG flake and validate their capability in identifying WC phases, we first apply them to the fractional quantum Hall regime at a large out-of-plane magnetic field ($B_\perp = 14$ T) in which a recent high-resolution STM study has unambiguously identified a WC phase[5]. At this field, a plethora of integer and fractional quantum Hall states (QHS) emerge. Notably, insulating behavior—characterized by a decrease in conductance with decreasing current—is not only observed within these QHS[32] but also observed close to the bandgap (Fig. 1b). In this latter regime, we observe a striking three-order-of-magnitude increase in the spectral current noise density at low DC current of 0.5 nA (Fig. 1c), consistent with the reported depinning of the magnetic-field-induced WC near the fractional filling factor −1/5, consistent with the STM measurement[5]. (These frequency-dependent measurements will be discussed in detail below.) This noise enhancement exhibits a strong dependence on both frequency and current, and the frequency-

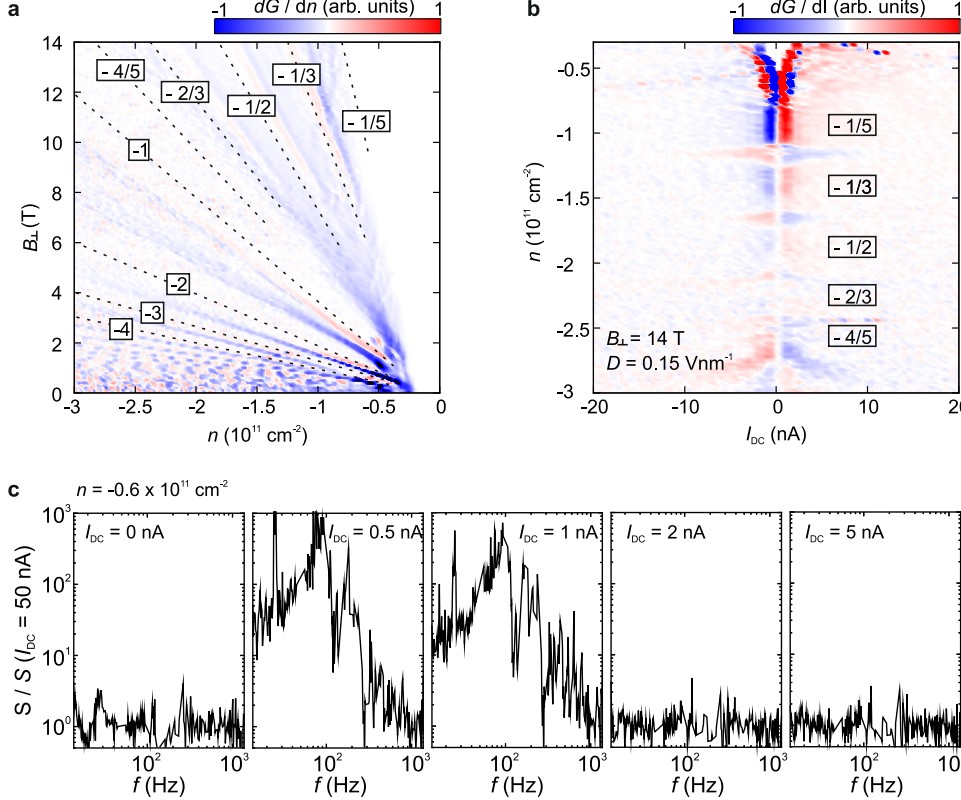

**Fig. 1 | Noise in the fractional quantum Hall regime of AB-stacked bilayer graphene. a** Derivative of the conductance (d$G$/d$n$) as a function of the charge carrier density $n$ and the out-of-plane magnetic field $B_\perp$ at an electric displacement field of $D = 0.15$ Vnm$^{-1}$. Integer and fractional quantum Hall states (QHS) emerge at finite $B_\perp$. Fractional QHS with filling factors $v < 1$ and integer QHS with $v \leq 4$ are traced by dashed lines. **b** d$G$/d$I$ as a function of the DC bias current $I_{DC}$ and the charge carrier density $n$. Insulating behavior (decreasing conductance with decreasing current) is

observed within the QHS[32] and close to the bandgap where a previous STM experiment revealed Wigner crystallization[5]. **c** Spectral noise density $S$ normalized with respect to $S$ ($I_{DC} = 50$ nA) as a function of the applied AC frequency $f$ ($I_{AC} = 100$ pA) measured at different $I_{DC}$, $n = -0.6 \times 10^{11}$ cm$^{-2}$, $B_\perp = 14$ T and $D = 0.15$ Vnm$^{-1}$. Frequency-dependent noise bulges appear for $I_{DC} = 0.5$ nA to $I_{DC} = 1$ nA. Data points that correspond to the noise floor of the measurement setup and the 50 Hz electrical grid were removed (see Methods).

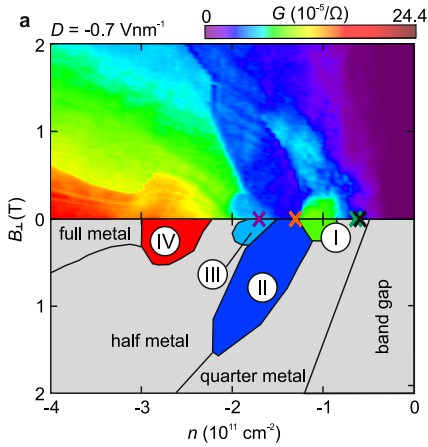
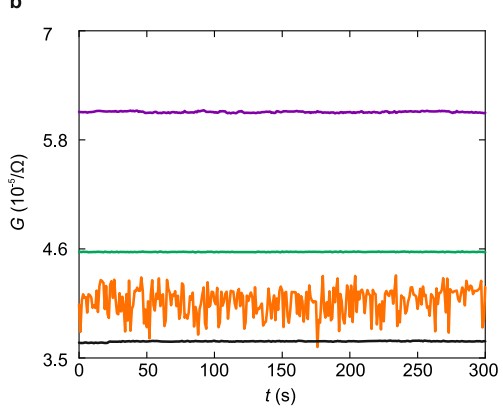

**Fig. 2 | Observation of noise in AB-stacked bilayer graphene at zero magnetic field. a** Conductance $G$ as a function of charge carrier density $n$ and out-of-plane magnetic field $B_\perp$ at an electric displacement field at $D = -0.7$ Vnm$^{-1}$ and an AC current (78 Hz) of 1 nA ($I_{DC} = 0$ nA). The various phases are schematically illustrated and labeled in the mirror image according to reference[23]. **b** Linecuts of the conductance as a function of time $t$ at zero magnetic field for the selected densities indicated by colored crosses in (**a**): $n = -1.7 \times 10^{11}$ cm$^{-2}$ (purple, phase III), $n = -1.3 \times 10^{11}$ cm$^{-2}$ (orange, phase II), $n = -0.65 \times 10^{11}$ cm$^{-2}$ (green, quarter metal), and $n = -0.6 \times 10^{11}$ cm$^{-2}$ (black, quarter metal).

dependent noise bulges appear at low DC currents up to 1 nA. Notably, the peak frequencies of the noise bulges increase with increasing $I_{DC}$ (see Supplementary Fig. 1). The disappearance of these noise bulges at higher currents ($I_{DC} > 1$ nA) indicates a transition away from the sliding regime, potentially due to a current-induced phase slip or competition with other competing quantum phases that have been shown to appear at low densities[33]. In contrast, no increase in the spectral current noise density can clearly be observed when $n$ is fine-tuned to the fractional quantum Hall regime with the filling factor $-1/3$ (see Supplementary Fig. 2).

Hereafter we shall examine the zero magnetic field regime with the same technique, where an unambiguous proof of a WC phase is still missing even though initial indications consistent with electron crystallization have emerged from bias spectroscopy and temperature-dependent resistance measurements[23,24]. We first reproduce the previous observations of the low-temperature phase diagram close to the WC phase to be studied. Figure 2a shows the measured two-terminal conductance map $G$ at $D = -0.7$ Vnm$^{-1}$ as a function of $n$ and $B_\perp$ measured with an applied AC current of 1 nA. At low $B_\perp$ and hole doping, a complex phase diagram emerges because of the interplay between electron-electron interaction and trigonal warping induced van Hove singularities[23,34–36], in the vicinity of which a variety of new correlated phases appear. Of special relevance for this work is an emergent phase in the vicinity of the fully isospin polarized van Hove singularity (phase II in Fig. 2a[23]) that manifests strong non-linearities with applied bias current and an insulating temperature dependence[23]. Both features are consistent with a WC phase. More attractively, this phase in the Landau fan diagram not only emanates from a range of finite densities at $B = 0$ but also follows the Středa formula with a quantized slope at finite $B_\perp$ fields[23]. Such topological phases, dubbed Wigner-Hall crystals[23] or anomalous Hall crystals[37–39], have sparked strong interest in the field. Here we aim to substantiate the WC nature of phase II.

A time-trace of the conductance within the density region of phase II is shown in Figs. 2b and 3c. Pronounced fluctuations in the conductance can be observed at an applied low-frequency (78 Hz) AC bias current of 1 nA, consistent with the sliding of a potential WC, as shown by the orange linecut in Fig. 2b. Crucially, we can exclude the possibility that these fluctuations arise from artifacts coming from contacts, as they manifest only within the density range associated with phase II (see the purple, black and green linecuts in Fig. 2b taken outside phase II with similar conductance). Furthermore, they become less pronounced in regions where phase II is destabilized by decreasing

$D$ (Supplementary Fig. 3) or increasing temperature (Supplementary Fig. 4). The fluctuations are negligible below AC bias currents of 100 pA (Supplementary Fig. 5), giving a first hint towards possible WC depinning and sliding behavior at larger AC biases to be investigated below. In the density region of the resistive phase III, the conductance fluctuations are much weaker (Fig. 2b, Supplementary Fig. 3). This phase also exhibits a much weaker dependence on temperature and current bias compared to phase II. Phase III is likely a potential WC phase with a much weaker pinning potential, and the depinning occurs at a much lower bias current[23]. Nevertheless, we shall focus on the further investigations of phase II.

To gain insights on the sliding behavior of the potential WC phase, we investigate the influence of an applied bias current on the resistance fluctuations (Figs. 3 and 4). We apply DC currents ($I_{DC}$) on top of a constant 78 Hz AC current ($I_{AC}$) of 100 pA at $D = -0.7$ Vnm$^{-1}$ and monitor the conductance (Fig. 3a, b) and its changes in the normalized resistance over time $\frac{dR}{dt} \times \frac{1}{R_0}$, with $R_0$ taken at time $t = 0$ s (Fig. 3c). There are no resistance fluctuations observed at low currents, indicative of a stable region where no depinning takes place. Despite that the conductance remains nearly constant up to $I_{DC} = 3.5$ nA within the density range of phase II (Fig. 3a, b), significant temporal fluctuations become evidenced at $I_{DC} \geq 1$ nA. Importantly, these fluctuations span the entire density range of phase II. This behavior is consistent with the depinning and sliding of a WC at applied DC currents of 1 nA to 5 nA. Above 4 nA, $\frac{dG}{dI_{DC}}$ exhibits a jump, which may indicate a phase slip induced by the large current. Notably, resistance fluctuations are also present at higher DC currents (~10 nA) near the phase boundary between phases II and phase I, and they can be attributed to the metal-insulator transition (Fig. 3c)[40].

The resistance fluctuations become increasingly pronounced when a small in-plane magnetic field $B_\parallel$ is applied, as exhibited in Fig. 4a–c. A small $B_\parallel$ can strengthen phase II[23] and consequently shift the threshold current to a higher value of $I_{DC}$. Intriguingly, resistance fluctuations are observed only at $I_{DC} \approx 5$ nA at $B_\parallel = 200$ mT (Fig. 4c), suggesting the onset of depinning and the onset of possible phase slip occur at nearly the same $I_{DC}$. More intriguingly at $B_\parallel > 300$ mT, no prominent resistance fluctuations are present and it appears that the possible phase slips could occur at even lower $I_{DC}$ values than the onset of depinning. These features deserve more elaborated examinations in the future.

In case of depinning and sliding WC's, the periodic WC lattice structure is anticipated to induce a periodic modulation of the current with a characteristic sliding frequency known as the washboard

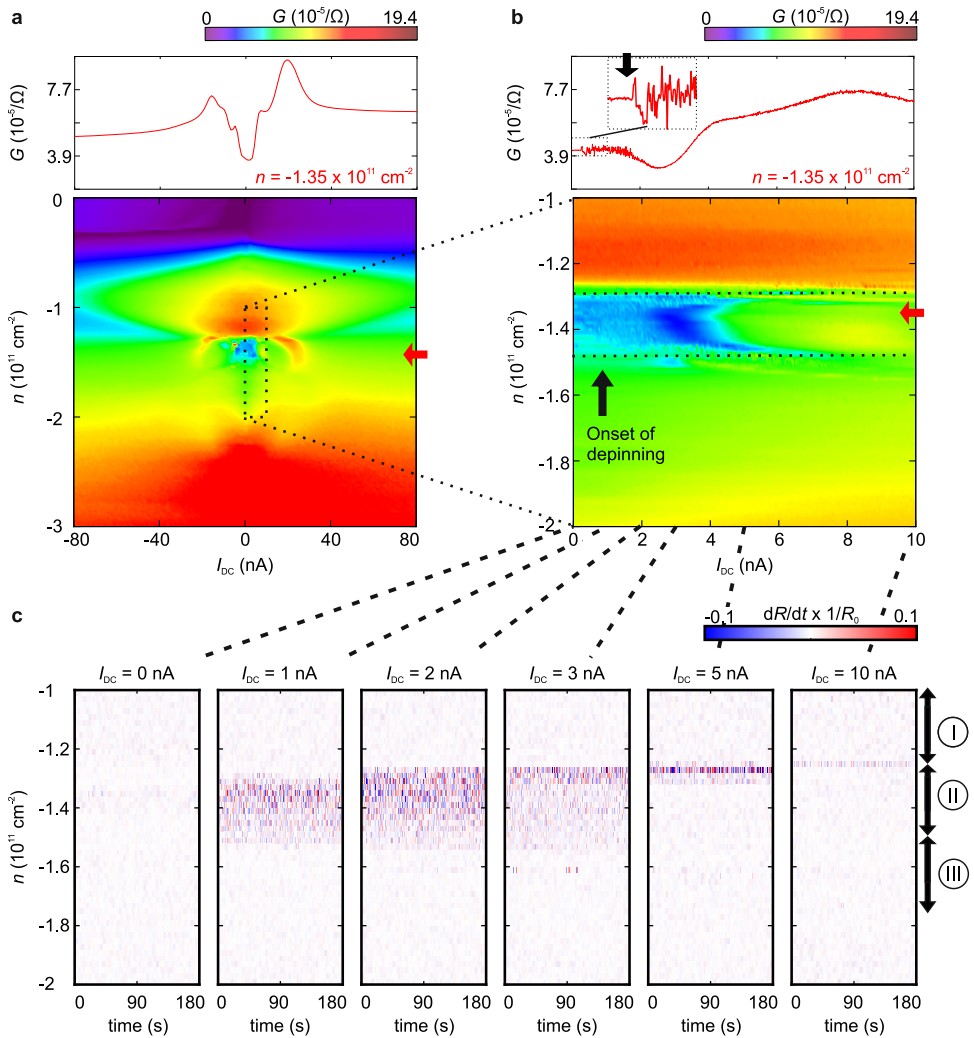

**Fig. 3 | Depinning as a function of the in-plane electric field. a** Conductance as a function of an applied DC bias current $I_{DC}$ and the charge carrier density $n$ at an electric displacement field $D = -0.7$ Vnm$^{-1}$ and an AC current $I_{AC} = 100$ pA. **b** Zoom-in of the density region of phase II in (**a**) (marked by dashed lines). Linecuts taken at $n = -1.35 \times 10^{11}$ cm$^{-2}$ (marked by red arrows) are shown in the top panel. Noise starts to appear at $I_{DC} \approx 200$ pA. The onset of conductance fluctuations is marked with a black arrow in the zoom-in. **c** Derivative of the normalized resistance over time $\frac{dR}{dt} \times \frac{1}{R_0}$ (the resistance $R$ is normalized with respect to $R_0$ taken at time $t = 0$ s) at different $I_{DC}$. The density region of phases I - II$I$ (determined at $I_{DC} = 0$ nA) is indicated by arrows.

frequency $f_0$, which is determined by the time-averaged velocity $v$ of the moving WC and the lattice constant $a_0$ of the WC: $f_0 = \frac{v}{a_0}$[6,10,12]. $a_0$ depends on $n$ via $a_0 = \sqrt{\frac{2}{\sqrt{3}n}}$, assuming a triangular lattice for the Wigner crystal[5]. $v$ may be tentatively given by $v = \frac{j}{en}$ with current density $j$ and electron charge $e$. Finally, the characteristic frequency at which the charge carriers' slide is given by $f_0 = \frac{v}{a_0} = \sqrt{\frac{\sqrt{3}}{2}} \frac{j}{e\sqrt{n}} = \sqrt{\frac{\sqrt{3}}{2}} \frac{I}{ew\sqrt{|n|}} \approx 5 \times 10^7$ Hz in our sample with sample width $w = 3$ μm, $I = 1$ nA, $|n| = 1.3 \times 10^{11}$ cm$^{-2}$ and $a_0 = 29.8$ nm.

To investigate the periodic modulation of the current and to experimentally determine $f_0$, we analyze the normalized spectral noise density $S$, i.e., a measure of the normalized variance of the measured voltage (see Methods). We employ an AC-DC interference technique, where a 100 pA AC signal with a tunable frequency $f$ is superimposed on the DC driving current $I_{DC}$. This approach serves as a sensitive probe of the washboard frequency, enabling the detection of DC-dependent resonances[15]. It is noteworthy that the measured frequency noise is usually much smaller (approximately three to four orders of magnitude[10,12]) than the expected f$_0$, and this is often explained by the motion of Wigner crystal domains (the systems is therefore sometimes

called a Wigner solid rather than a Wigner crystal) and its interaction with the disorder potential rather than the motion of an entire, defect free crystal[10,12]. Therefore, although we expect $f_0$ - 10 MHz at $n = -1.3 \times 10^{11}$ cm$^{-2}$ and I = 1 nA, we restricted the frequency range to $f < 1370$ Hz (electrical filters integrated into our cryostat cut out high frequencies), and within this range $G$ remains nearly constant (Supplementary Fig. 6).

In the absence of sliding, i.e., at $I_{DC} = 0$ nA and $I_{DC} > 5$ nA (Fig. 5a), and outside the density range of phase II (Supplementary Fig. 6c), $S$ is predominantly characterized by a $1/f$ or "flicker" noise spectrum, as shown by the linecuts for $I_{DC} = 0$ nA, $I_{DC} = 20$ nA and $I_{DC} = 80$ nA in Fig. 5a. This type of noise spectrum is commonly observed across a frequency range spanning from a few Hz to tens of kHz in various materials and arises from different fluctuation processes, such as mobility fluctuations caused by scattering centers within the substrate or sample[41].

Deviations from the $1/f$ noise spectrum appear in phase II within its specific ranges of $I_{DC}$ and $n$, where fluctuations in $R$ over time are evident (Figs. 3, 4, 5). In this regime, $S$ exhibits a significant increase by orders of magnitude (Fig. 5a). To separate the enhanced noise spectrum from the underlying $1/f$ background noise, Fig. 5b displays the normalized spectral noise density, i.e., $S$ divided by $S$ ($I_{DC} = 80$ nA), as a

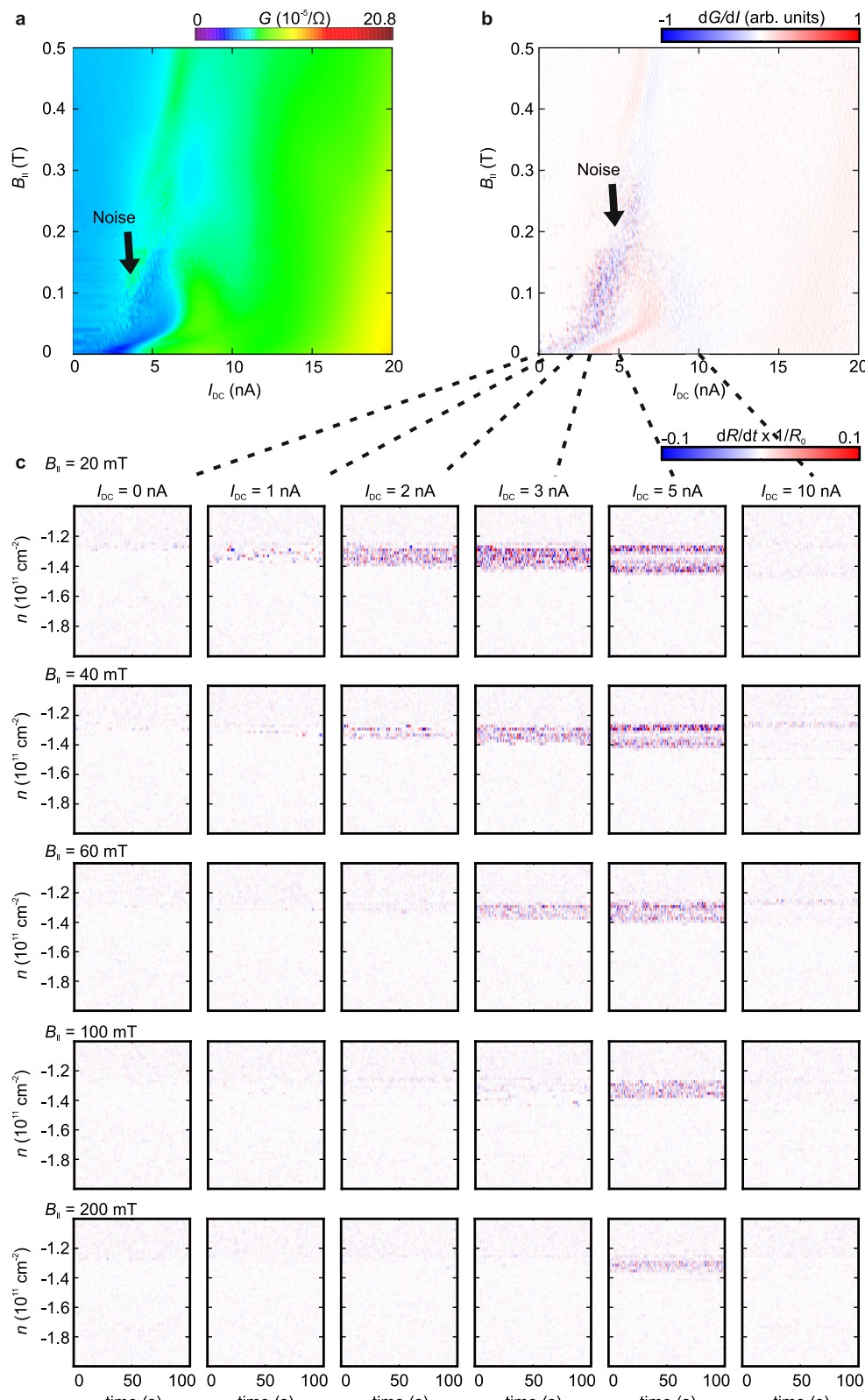

**Fig. 4 | Noise as a function of the in-plane magnetic field.** Conductance ($G$) (**a**) and derivative of the conductance with respect to the applied DC current $I_{DC}$ (d$G$/d$I$) (**b**) as a function of $I_{DC}$ and an in-plane magnetic field $B_{\parallel}$ taken at a charge carrier density $n = -1.3 \times 10^{11}$ cm$^{-2}$ and an electric displacement field $D = -0.7$ Vnm$^{-1}$. Current fluctuations, labeled as noise, emerge at $B_{\parallel} < 0.3$ T and $I_{DC} < 5$ nA before the Wigner crystal/solid is fully depinned at lager $I_{DC}$. **c** Derivative of the normalized resistance over time $\frac{dR}{dt} \times \frac{1}{R_0}$ as a function of the time $t$ and $n$ at different $I_{DC}$ and $B_{\parallel}$.

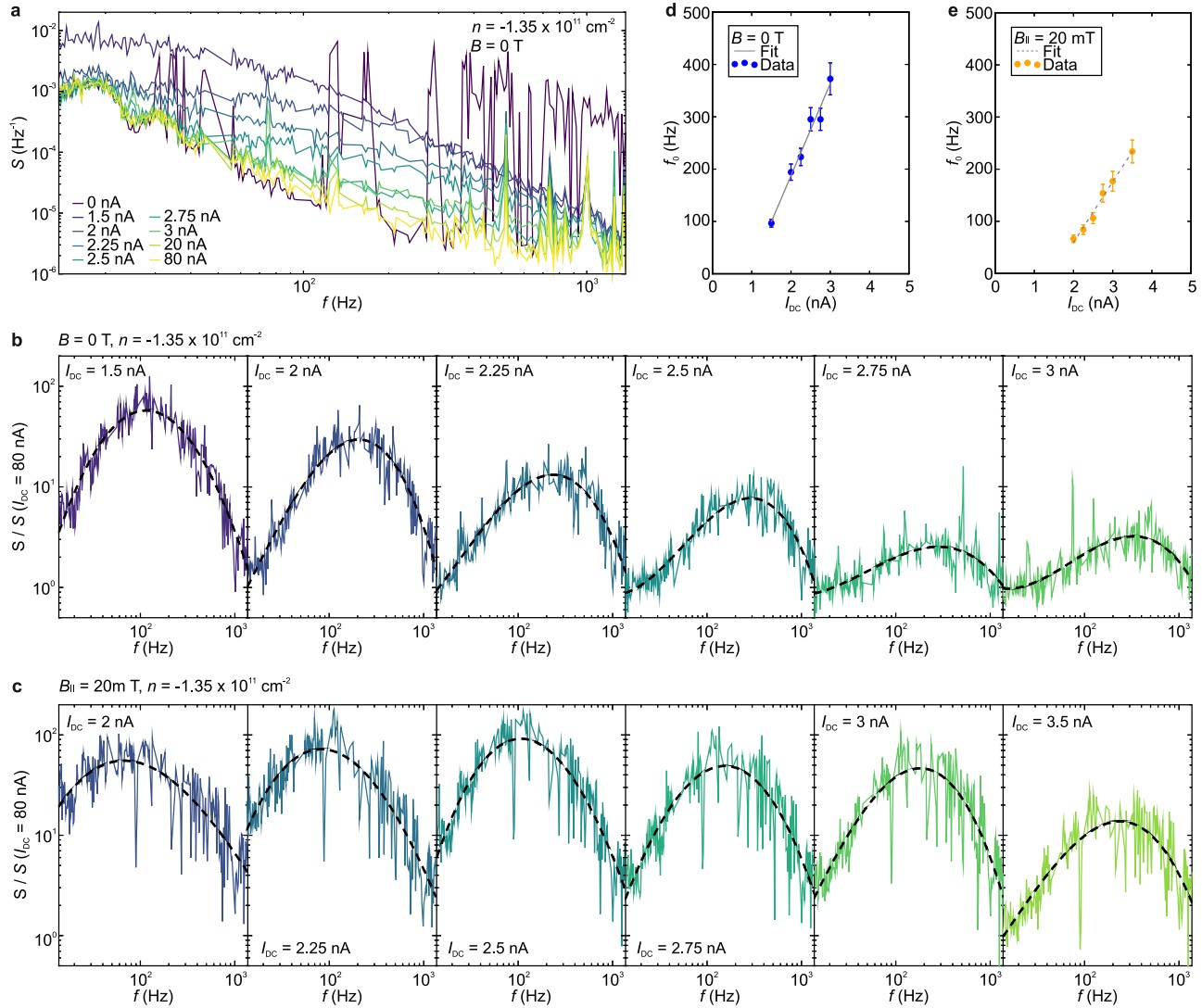

**Fig. 5 | Frequency dependence of the current fluctuations. a** Normalized spectral noise density $S$ as a function of the applied frequency $f$ of the AC current $I_{AC}$ at different $I_{DC}$ at zero magnetic field $B$, an electric displacement field $D = -0.7$ Vnm$^{-1}$ and a charge carrier density of $n = -1.35 \times 10^{11}$ cm$^{-2}$ within phase II. Data points that correspond to the noise floor of the measurement setup and the 50 Hz electrical grid were removed (see Methods). Spectral noise density normalized relative to the background spectral noise density measured at $I_{DC} = 80$ nA ($S / S (I_{DC} = 80$ nA)) at a function of $f$ at different $I_{DC}$ and an in-plane magnetic field $B_{II}$ of 0 T (**b**) and 20 mT (**c**). The noise bulges are fitted with polynomial functions represented by dashes lines. Dependence of the washboard frequency $f_0$ at $B_{II} = 0$ T (**d**) and $B_{II} = 20$ mT (**e**) extracted from (**b**, **c**) as a function of $I_{DC}$. Error bars represent the uncertainty in the extracted peak frequency from the polynomial fit. Linear fits are shown in gray.

function of $f$ for different $I_{DC}$. The appearance of noise bulges, each centered around a characteristic frequency $f_0$, and their evolution with increasing $I_{DC}$ are clearly visible. The noise bulges exhibit an inhomogeneous broadening, likely due to the presence of competing processes with different time constants[10]. To determine $f_0$, we apply polynomial fits to the different spectra (note that Lorentzian noise spectra would be expected without the presence of inhomogeneous broadening[10]). Figure 5d illustrates the trend of $f_0$ as a function of $I_{DC}$ for $B_{II} = 0$ T and $B_{II} = 20$ mT. Consistent with the depinning and sliding behavior of WCs or Wigner solids, $f_0$ increases linearly with $I_{DC}$ with and without the applied $B_{II}$, providing another compelling transport signature for the formation of a WC or Wigner solid in phase II. Note that phase II exhibits in-plane spin polarization[23]. As a result, applying an in-plane magnetic field stabilizes the Wigner phase, leading to the onset of sliding at higher current levels (Figs. 4, 5c). Furthermore, the application of an in-plane magnetic field may alter the domain configuration within the sample. Both effects can result in the different slopes seen in Fig. 5d, e.

Our measured $f_0$ values are consistent with that of a WC in a GaAs quantum well[12]. Similar to this well-established case, if the time-averaged velocity is interpreted as that of the moving WC, the estimated $f_0$ is five orders of magnitude larger. This mismatch was similarly present in the GaAs quantum well and explained by the dynamics of domains. Our case is more subtle and cannot be solely explained by the formation of domains, thereby calling for future theoretical and experiment inspections. Specifically, we extract the threshold electric field and then estimate the domain size in the presence of disorder to be $L_0 \sim 2$ µm (by using $L_0 = \mathrm{sqrt}\left(\frac{0.02e}{4\pi\varepsilon\varepsilon_0 E_{th}}\right)$), with threshold electric field $E_{th} = \frac{I_{th}}{l^*G} = 2.5 V/m$, threshold current $I_{th} = 1$ nA, sample length $l = 10 \, \mu m$ and conductance $G = 4 \times 10^{-5} \, 1/\Omega$[12,42]. We estimate the expected frequency for a sliding mode crossing a single domain (f = v/$L_0$) to be around 320 kHz, which is still substantially higher than the observed ~100 Hz. Other possible explanations for the discrepancies in frequency include variations in the current distribution and thus in the in-plane electric field (due to contact effects), thermal fluctuations, and overestimation of the average velocity.

Finally it is noteworthy that we identified similar signatures of WC depinning and sliding also in electron-doped AB-stacked BLG[24] (Supplementary Fig. 7) using the same bias-dependent noise measurements applied here to the hole-doped regime. Additionally, we note that in other AB-stacked BLG experiments no clear signs consistent with WC have been found[35]. In a similar region of the phase diagram, but at higher temperatures and over a broader density range, the formation of magnetic domains has been observed[35]. While it remains to be investigated whether such domains could produce the type of characteristic noise detected in our experiments, it seems unlikely since at least in noise studies of spin transfer torque random telegraph noise with a 1/f spectrum has been observed[43–47] – at odds with the low-frequency noise observed for depinning of WC domains. Potentially the ground state of AB-stacked BLG depends on the strength of the exchange interaction, which is screened by nearby metallic gates, as well as subtle variations in sample quality in this region of the phase diagram.

In conclusion, the analysis of the noise spectra aligns with previous experimental and theoretical studies on the depinning and sliding WC[7–15,17–20,22,41], consistent with the interpretation that the low-density, insulating phase II observed in AB-stacked BLG at zero magnetic field, indeed, originates from the formation of a Wigner crystal or solid. An out-of-plane or in-plane magnetic field can strengthen the Wigner phase, increase the depinning onset, and shift the threshold current to a higher value that eventually becomes unobservable beyond a critical field (Fig. 4 and Supplementary Fig. 8). A particular intriguing feature of phase II (Figs. 2–5) is its dependence on the out-of-plane magnetic field, which exhibits consistency with a low-density quantum anomalous Hall state with a Chern number of 2[23]. This underscores the need for alternative methods to verify or dispute the topologically non-trivial nature of phase II in AB-stacked BLG. It is expected that WC phases may also arise in thicker graphene systems with rhombohedral stacking because of the electric gate tunability of the ultra-flat bands near their band gaps at charge neutrality.

## Methods
### Electrical measurements
All electrical measurements were conducted using a standard lock-in technique. An AC reference signal with a frequency of 78 Hz (unless otherwise specified) was generated by a lock-in amplifier (SR865A, Stanford Research Systems) and converted into a small AC current signal of 100 pA–1 nA via a high resistor. The voltage drop across the contacts was then measured using a second lock-in amplifier. Note that we did not account for finite contact resistances, nor did we subtract any contact resistance from our conductance values. An additional DC bias current was introduced using a DC source-measure unit (SourceMeter 2450, Keithley) and a resistor. This DC bias was modulated onto the AC reference signal via a home build transformer. To prevent grounding loops, the measurement units were isolated from standard power lines using isolating transformers.

We can independently tune the charge carrier density $n$ and the electric displacement field $D$ using graphite top and bottom gates. The charge carrier density $n$ is defined as $n = \varepsilon_0 \varepsilon_{hBN}(V_t/d_t + V_b/d_b)/e$, where $V_t$ and $V_b$ are the gate voltages applied to the top and bottom gates, respectively, $d_t$ and $d_b$ are the thicknesses of the upper and lower hBN flakes serving as dielectrics., $e$ is the charge of an electron, $\varepsilon_{hBN}$ is the dielectric constant of hBN, and $\varepsilon_0$ is the vacuum permitivity. The vertical electric displacement field $D$ is defined as $D = \varepsilon_{hBN}(V_t/d_t - V_b/d_b)/2$.

### Noise measurements at nA current levels
Noise spectroscopy has been effectively utilized to detect charge density wave (CDW) and Wigner crystal sliding in various material systems. In materials such as $TaS_2$, measurements are typically conducted at higher current levels, often in the mA range. For instance,

Mohammadzadeh et al.[10] reported on such measurements in $TaS_2$. In contrast, recent studies on high-mobility GaAs heterostructures[12], SiGe/Si/SiGe heterostructures[48] and Si MOSFETs[11] have demonstrated that noise measurements at nanoampere current levels can successfully probe Wigner solid behavior. This illustrates the adaptability of noise spectroscopy across different current regimes, depending on the material system under investigation.

In experiments with bilayer graphene heterostructures, nA current levels are standard when probing correlated electronic phases. Specifically, we observe that the fractional quantum Hall states, as shown in Fig. 1b of our manuscript, disappear at currents of just a few nA.

To detect the noise signals associated with nA currents, the normalized spectral noise density $S$, which is defined as $S = v^2/(ENBW \times V_{Contacts}^2)$, is detected using the same lock-in amplifier (SR865A, Stanford Research Systems) used to measure the voltage drop across the contacts. In the definition of $S$, $v$ represents the variance determined from the lock-in amplifier, $ENBW = 0.78$ Hz is the equivalent noise bandwidth, and $V_{Contacts}$ is the voltage drop between the contacts. $v$ was determined using an integration time of 20 s, corresponding to at least 200 data points for each chosen frequency.

### Evaluation of the frequency-dependent data
The frequency dependent data shown in Supplementary Fig. 6 is impacted by additional sharp peaks that occur at approximately 50 Hz and its higher harmonics, corresponding to electrical power distribution noise (the standard mains frequency in Germany is 50 Hz). To improve the data presentation and to extract the washboard frequency $f_0$, we have removed data points that correspond to the noise floor of the measurement setup and the 50 Hz electrical grid[9].

## Data availability
Relevant data supporting the key findings of this study are provided within the article. All raw, processed, and fitting datasets used to generate the figures and perform the analysis are available from the corresponding authors upon request.

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

## Acknowledgements

We acknowledge illuminating discussions with E.Y. Andrei, Shafayat Hossain, A. Gosh, A.F. Young and D. Steil. R.T.W. and A.M.S acknowledge funding from the Deutsche Forschungsgemeinschaft (DFG, German Research Foundation) under the SFB 1073 project B10. R.T.W. acknowledges funding from the DFG SPP 2244. K.W. and T.T. acknowledge support from the JSPS KAKENHI (Grant Numbers 21H05233 and 23H02052) and World Premier International Research Center Initiative (WPI), MEXT, Japan. F.Z. acknowledges supports from US National Science Foundation under grants DMR-1945351, DMR-2324033, and DMR-2414726.

## Author contributions

A.M.S. fabricated the device. A.M.S. and M.S. conducted the measurements with assistance from C.E., I.W., and J.P. A.M.S. performed the data analysis with help from M.S., C.E., I.W., and J.P. K.W. and T.T. grew the hexagonal boron nitride crystals. All authors discussed and interpreted the data. R.T.W. supervised the experiments and the analysis. A.M.S.,

F.Z., and R.T.W. prepared the manuscript with input from M.S., C.E., I.W., and J.P. and feedback from all authors.

## Funding

## Competing interests
The authors declare no competing interests.
