## [Transparent Peer Review file · Nature Communications]

Signatures of sliding Wigner crystals in bilayer graphene at zero and finite magnetic fields

Corresponding Author: Professor R. Thomas Weitz

Version 0:

Reviewer comments:

Reviewer #1

(Remarks to the Author)

The authors report on signatures of sliding Wigner crystals in bilayer graphene at zero and finite magnetic fields. They use the "noise spectroscopy" methods (measurements of low-frequency current fluctuations and derivative IVs) to monitor the effects. The paper is interesting and timely, given the renewed interest to the CDWs (Wigner crystals) and the emerging noise spectroscopy approaches.

I have one technical question: How reliable are noise measurements at nA current levels? Discussion of this would be beneficial for the readers.

Reviewer #2

(Remarks to the Author)

Seiler et al. perform transport measurements in correlated states in AB-stacked bilayer graphene devices. Focusing on the symmetry-broken correlated insulator at high displacement fields, the authors report conductance fluctuations with interesting behavior as a function of current bias, magnetic field, and carrier density. Notably, the noise amplitude shows a peak when the frequency of the AC current bias is tuned between 100 Hz and 700 Hz. The peak frequency depends linearly on the applied DC current bias. These signatures are interpreted as appearing from a sliding/depinning of a Wigner crystal.

The theme of the paper is topical. Zero-magnetic field Wigner crystal ground states have indeed been predicted to arise in bilayer and multilayer graphene. Furthermore, in the case of the present work, the putative Wigner crystal state appears simultaneously with a Chern number of 1, indicating a rather exotic crystalline solid. However, I am not convinced that the noise signatures can be uniquely attributed to a Wigner crystal.

While the authors draw parallels with previous experiments where "similar" noise signatures have been observed for Wigner crystal states in GaAs quantum wells, it must be noted that Wigner crystals in these systems have been definitively identified by other methods, such as microwave spectroscopy. The issue becomes more complicated when one considers that based on noise measurements alone, it is hard to distinguish a "crystal" of electrons from "amorphous" states of electrons, such as an electron glass or a Wigner solid.

Here are some comments that may help address these issues:

1. Starting with the "calibration" experiment. The authors show that there is a noise bulge for a doping density of $n = 0.6 \times 10^{11} \text{ cm}^{-2}$ where a Wigner crystal is expected. Can the authors show what happens for other carrier densities where the sample is insulating but is NOT expected to have a Wigner crystal ground state, such as a fractional Hall state at $\nu = -1/5$ or maybe even an integer Hall state at $\nu = 1$.
2. It is well known that metal-insulator transitions produce large noise when the transition is approached (See Jaroszyński, J., Dragana Popović, and T. M. Klapwijk. "Universal Behavior of the Resistance Noise across the Metal-Insulator Transition in Silicon Inversion Layers." *Physical review letters* 89.27 (2002): 276401, for example). Have the authors considered such an interpretation? I am not convinced that the noise enhancement alone "provides compelling evidence for the presence and the depinning of the magnetic field induced WC ...".

3. Can the authors show that the peak frequency of the noise bulge also scales linearly with I_{dc} in the calibration experiment performed in the quantum Hall regime ($n=0.6 \times 10^{11}/\text{cm}^2$)? What happens at other insulating states that are not Wigner crystals?

4. The data in Fig. 5, which purportedly provides the strongest signature of the Wigner crystal phase is not very high quality. The bulge is not visible in the raw data and appears only after normalization. This is unlike the GaAs experiments, where the noise peak is well-defined and does not need a background correction. At the very least, error bars should be provided for Figs. 5d and 5e, considering how broad these features are.

5. Can the authors convincingly rule out contact effects? Insulating states in graphene are particularly prone to various transport artifacts because of contact equilibration issues. It would be important to show that the noise signatures don't arise from the contacts.

In light of these issues, I cannot recommend publication in the present form of the manuscript.

Reviewer #3

(Remarks to the Author)

Seiler et al. report on studies of sliding Wigner crystals (WC) in the bilayer graphene system at zero and high magnetic fields. They primarily do so by revealing non-linear IV characteristics and noise in the posited WC regime of AB-stacked bilayer graphene.

Comments:

The authors first show data in the high field regime, but I find the discussion of the data, which is restricted to one paragraph, inadequate. For example, the authors cite Ref. [4] as direct evidence for the presence of WC. In Ref. [4], the presence of WC is reported in a filling range $0.13 < \nu < 0.38$. However, the authors here find non-linear IV signatures only at $\nu < 0.2$. Why are the non-linear IV signatures not seen at higher filling factors? Further, the noise generated due to sliding vanishes by 2 nA. Why does this happen in the high field regime? The authors report the same behavior in the zero field case, and attribute it to phase slippage due to a large current. They should elaborate on this. Why is there a discrepancy in the maximum current needed for this phase slip at high (2 nA) and the zero field (4-5 nA) case?

The conductances are in arbitrary units. The authors should show the exact values in the figures [for example Fig. 3(a)]. Also, the authors should estimate the related threshold electric fields based on the conductance and the threshold current, and then estimate the domain size [based, e.g., on Phys. Rev. B 17, 535 (1978)]

In Fig. 3(b), the conductance falls even further (about 25%) past the conductance fluctuation regime. If this is past the phase slip regime due to large currents, it is unclear why the conductance has to drop even lower than the pinned phase at very low I_{DC} .

The discrepancies associated with the washboard frequencies (~ 100 Hz and ~ 10 MHz), if attributed to domain sizes, would correspond to domains of the order $L = \left[\frac{10^5 \text{ a}}{f} \right] \approx 0$. ($f \sim \nu/L$ instead of ν/a_0) which far exceeds the sample dimensions (5 μm), implying the presence of a single domain within the sample. But is this really the case? The authors should discuss this. Also, from Fig. 5(b,c), it can be gathered that the integrated noise in the very low-frequency regime continuously decreases. Can the authors comment on this, since I would expect the sliding noise to be nearly constant. Also, the differing slopes in Fig. 5(d) and 5(e) suggest that the domain sizes are different at different parallel fields. The authors should elaborate on why the domain size depends on parallel field.

General comment: I find, in general, the figure quality poor, especially the ones which show the conductance fluctuations. These must be improved.

Summary: Although the authors show some potentially interesting data, I am not convinced that the observations provide clear evidence for sliding WC states. Hence, I cannot recommend publication.

Reviewer #4

(Remarks to the Author)

Version 1:

Reviewer comments:

Reviewer #1

(Remarks to the Author)

The authors addressed the questions. I recommend acceptance.

Reviewer #2

(Remarks to the Author)

I thank the authors for carefully answering all questions that I have raised. The answers are satisfactory, and it clears the bar for publication. The nature of insulating ground states in bilayer and multilayer graphene remains mysterious, and probes beyond conventional transport are urgently needed. I congratulate the authors for a timely and interesting experiment that the community will benefit from.

Reviewer #3

(Remarks to the Author)

The authors present additional data in the high-field regime; however, in examining Extended Figure 1, the frequency peak appears poorly resolved, and the quality of the fit is suboptimal (likely due to the apparent bimodality of the observed signal near ~ 20 Hz and ~ 100 Hz). To me, there appears to be minimal dependence of the peak frequency on the applied current ($f \approx 100$ Hz for all drives), raising questions about whether the signal can be attributed to WC dynamics.

In response to concerns about domain size estimates, the authors suggest alternative mechanisms for the observed frequencies including variations in the current distribution and thus in the in-plane electric field (due to contact effects), thermal fluctuations, and overestimation of the average velocity. While I understand it is hard to quantify the contribution of these effects to the data, taking into account the broad narrow band noise peaks, the vanishing of the noise peaks past a threshold drive, one wonders if these contributions are significant.

However, in terms of the raw data itself, there is consistency between the high and the low field data (though I find the quality to be poor and question whether the whole phenomena can be attributed to WC dynamics at all). While much remains to be understood, I feel the new set of data provided by Seiler et al. is novel and could add on to the repository and further understand general dynamic responses of CDWs to an electric field.

Reviewer #4

(Remarks to the Author)

REVIEWER COMMENTS

Reviewer #1 (Remarks to the Author):

The authors report on signatures of sliding Wigner crystals in bilayer graphene at zero and finite magnetic fields. They use the "noise spectroscopy" methods (measurements of low-frequency current fluctuations and derivative IVs) to monitor the effects. The paper is interesting and timely, given the renewed interest to the CDWs (Wigner crystals) and the emerging noise spectroscopy approaches.

Reply:

We thank the referee for carefully reading our manuscript. We are pleased that the referee found our work both interesting and timely.

I have one technical question: How reliable are noise measurements at nA current levels? Discussion of this would be beneficial for the readers.

Reply:

We appreciate the referee question regarding the reliability of noise measurements at nA current levels.

Noise spectroscopy has been effectively utilized to detect the sliding of charge density wave (CDW) and Wigner crystal in various material systems. In materials such as TaS₂ (Mohammadzadeh et al., Appl. Phys. Lett. 118 (2021)), measurements were typically conducted at higher current levels, often in the mA range. In contrast, recent studies on high-mobility GaAs heterostructures (Madathil et al., Phys. Rev. Lett. 131 (2023)), SiGe/Si/SiGe heterostructures (Melnikov et al., Phys. Rev. B 109 (2024)) and Si MOSFETs (Brussarski et al., Nat. Commun. 9, 3803 (2018)) have demonstrated that noise measurements at nA current levels can successfully probe Wigner crystal behavior. This illustrates the adaptability of noise spectroscopy across different current regimes, depending on the material systems under investigation.

In our experiments with bilayer graphene heterostructures, nA current levels are standard when probing correlated electronic phases. Specifically, we have observed that the fractional quantum Hall states, as shown in Fig. 1b of our manuscript, disappear at currents of just a few nA. This behavior aligns with previous studies, indicating that nA current levels are appropriate for the present study.

To detect the small signals associated with nA currents, all electrical measurements were conducted using a standard lock-in technique. An AC reference signal with a frequency of 78 Hz (unless otherwise specified) was generated by a lock-in amplifier (SR865, Stanford Research Systems) and converted into a small AC current signal of 100 pA to 1 nA via a resistor of high resistance. The voltage drop across the contacts was then measured using a second lock-in amplifier capable of measuring signals below a few nanovolts. The normalized spectral noise density S was determined using the same lock-in amplifier.

To further validate our approach of using noise measurements to probe Wigner crystal states at nA current levels, we have performed a calibration experiment in the fractional quantum Hall regime, where Wigner crystal states in between two fractional states have

been independently confirmed by a scanning tunneling microscopy study (Y.-C. Tsui et al. (ref. 5)). The consistency between our noise measurements and prior STM results reinforces the credibility of our technique.

To address the referee question, we have added the discussion on the detection of noise at nA current levels to the methods section of the revised manuscript. Furthermore, we have extended the section on our calibration experiment in the fractional quantum Hall regime in the main text.

Reviewer #2 (Remarks to the Author):

Seiler et al. perform transport measurements in correlated states in AB-stacked bilayer graphene devices. Focusing on the symmetry-broken correlated insulator at high displacement fields, the authors report conductance fluctuations with interesting behavior as a function of current bias, magnetic field, and carrier density. Notably, the noise amplitude shows a peak when the frequency of the AC current bias is tuned between 100 Hz and 700 Hz. The peak frequency depends linearly on the applied DC current bias. These signatures are interpreted as appearing from a sliding/depinning of a Wigner crystal.

Reply:

We appreciate the referee for carefully reading our manuscript and for summarizing our work.

The theme of the paper is topical. Zero-magnetic field Wigner crystal ground states have indeed been predicted to arise in bilayer and multilayer graphene. Furthermore, in the case of the present work, the putative Wigner crystal state appears simultaneously with a Chern number of 1, indicating a rather exotic crystalline solid. However, I am not convinced that the noise signatures can be uniquely attributed to a Wigner crystal.

While the authors draw parallels with previous experiments where “similar” noise signatures have been observed for Wigner crystal states in GaAs quantum wells, it must be noted that Wigner crystals in these systems have been definitively identified by other methods, such as microwave spectroscopy. The issue becomes more complicated when one considers that based on noise measurements alone, it is hard to distinguish a “crystal” of electrons from “amorphous” states of electrons, such as an electron glass or a Wigner solid.

Reply:

We thank the referee for providing insightful feedbacks on our work and for considering our study topical. We also appreciate the referee’s concerns regarding the attribution of the observed noise signatures to a Wigner crystal state. *In particular, we agree that it is challenging to distinguish Wigner crystal and solid states by noise measurements. To address this, we have added the possibility of a Wigner solid in our revised manuscript.* We also recognize the importance of complementary techniques, such as microwave spectroscopy or scanning tunneling microscopy, in order to conclusively differentiate between these two forms of electronic ordering. On the other hand, we note that other transport signatures have been shown in ref. 23, and that more and more techniques will be applied to this “two-year old” (*by the submission date of this current work*) state in the future.

While increased noise levels are observed in amorphous electron glasses, too, these systems lack the characteristic – the dependence of noise peak frequency on the applied DC current bias – that we have observed in our experiments. This feature is discussed in greater detail in our response to comment #2.

To address the referee’s concerns, we have made several revisions in the manuscript. These revisions are detailed below and highlighted in blue for clarity.

Here are some comments that may help address these issues:

1. Starting with the “calibration” experiment. The authors show that there is a noise bulge for a doping density of $n = -0.6 \times 10^{11} \text{ cm}^{-2}$ where a Wigner crystal is expected. Can the authors show what happens for other carrier densities where the sample is insulating but is NOT expected to have a Wigner crystal ground state, such as a fractional Hall state at $\nu = -1/5$ or maybe even an integer Hall state at $\nu = 1$.

Reply:

We thank the referee for this valuable suggestion. We have performed additional “calibration” experiments within the fractional quantum Hall state at $\nu = -1/3$, which corresponds to $n = -1.4 \times 10^{11} \text{ cm}^{-2}$ at $B = 14 \text{ T}$. The results have been added to the revised Extended Data (Extended Data Fig. 2), as well as reproduced in Fig. R1 below. As expected, we do not observe an increased noise level with increasing applied current here. Moreover, at $I_{DC} = 50 \text{ nA}$ (the data at $I_{DC} = 50 \text{ nA}$ is used for normalization), the *background* spectral noise density at $n = -1.4 \times 10^{11} \text{ cm}^{-2}$ (fractional quantum Hall state) is at the same level as that at $n = -0.6 \times 10^{11} \text{ cm}^{-2}$ (Wigner phase), as seen in Fig. R1b.

Fig. R1. Frequency dependence of the spectral noise density within a fractional quantum Hall state. (a) Normalized spectral noise density S relative to the background spectral noise density measured at $I_{DC} = 50 \text{ nA}$ as a function of the applied frequency f of the AC current I_{AC} at different I_{DC} at an out-of-plane magnetic field $B = 14 \text{ T}$, an electric displacement field $D = 0.15 \text{ V nm}^{-1}$ and a charge carrier density of $n = -1.4 \times 10^{11} \text{ cm}^{-2}$ (fractional QHS with filling factor $\nu = -1/3$). S does not show any frequency dependence or any particular features. (b) S relative to the background spectral noise density measured at $n = -0.6 \times 10^{11} \text{ cm}^{-2}$ (see Fig. 1 for more data taken at $n = -0.6 \times 10^{11} \text{ cm}^{-2}$) as a function of f .

2. It is well known that metal-insulator transitions produce large noise when the transition is approached (See Jaroszyński, J., Dragana Popović, and T. M. Klapwijk. "Universal Behavior of the Resistance Noise across the Metal-Insulator Transition in Silicon Inversion Layers."

Physical review letters 89.27 (2002): 276401, for example). Have the authors considered such an interpretation? I am not convinced that the noise enhancement alone “provides compelling evidence for the presence and the depinning of the magnetic field induced WC ...”.

Reply:

We sincerely appreciate the referee’s insightful question regarding the possible connection between the observed noise enhancement and metal-insulator transitions. While it is well established that noise can be significantly amplified near a metal-insulator transition, we emphasize that the enhanced noise spectral density that we report emerges within phase II rather than at the phase boundary. Specifically, the metal-insulator transition in our system occurs at $n = 1.25 \times 10^{11} \text{ cm}^{-2}$ (for $D = -0.7 \text{ V/nm}$ and $B = 0 \text{ T}$), whereas we observe a pronounced noise enhancement over the density range $1.25 \times 10^{11} \text{ cm}^{-2} < n < 1.5 \times 10^{11} \text{ cm}^{-2}$ (see Fig. R2 below). Also, the noise data was measured at 10 mK, far below the critical temperatures.

Indeed, we do observe fluctuations in conductance at much higher currents ($I_{DC} = 10 \text{ nA}$) near the phase boundary, which could be linked to the metal-insulator transition. While it remains possible that similar fluctuations exist at lower currents, our data, especially the frequency-dependent data shown in Fig. 5, indicate that the primary noise enhancement at low I_{DC} is irrelevant to metal-insulator transitions.

To address this important point raised by the referee, we have added a discussion of the noise at the metal-insulator transitions to our manuscript.

Fig. R2. Depinning as a function of the charge carrier density and in-plane electric field. (a) Conductance as a function of an applied DC bias current I_{DC} and the charge carrier density

n at an electric displacement field $D = -0.7 \text{ Vnm}^{-1}$ and an AC current $I_{AC} = 100 \text{ pA}$. **(b)** Zoom-in of the density region of phase II in **(a)** (marked by dashed lines). Line cuts taken at $n = -1.35 \times 10^{11} \text{ cm}^{-2}$ (marked by red arrow) are shown in the top panel. Noise starts to appear at $I_{DC} \approx 200 \text{ pA}$. The onset of conductance fluctuations is marked with a black arrow in the zoom-in. **(c)** Derivative of the normalized resistance over time $dR/dt \times 1/R_0$ (the resistance R is normalized with respect to R_0 taken at time $t = 0 \text{ s}$) at different I_{DC} . The density region of phases I - III (determined at $I_{DC} = 0 \text{ nA}$) is indicated by arrows.

3. Can the authors show that the peak frequency of the noise bulge also scales linearly with I_{DC} in the calibration experiment performed in the quantum Hall regime ($n = -0.6 \times 10^{11} \text{ cm}^{-2}$)? What happens at other insulating states that are not Wigner crystals?

Reply:

We thank the referee for this question regarding the peak frequency scaling of noise bulges. We have conducted additional measurements to further investigate this behavior. These measurements are shown below (Fig. R3). Indeed, we observe that at $n = -0.6 \times 10^{11} \text{ cm}^{-2}$ ($B_{\perp} = 14 \text{ T}$ and $D = 0.15 \text{ Vnm}^{-1}$) the peak frequencies of the noise bulges increase with increasing I_{DC} . We have added Fig. R3 as Extended Data Fig. 1 to the revised version of the manuscript.

Fig. R3. Frequency dependence of the spectral noise density within the magnetic-field induced Wigner crystal state. (a) Spectral noise density S normalized with respect to $S(I_{DC} = 0 \text{ nA})$ as a function of the applied AC frequency f ($I_{AC} = 100 \text{ pA}$) measured at different I_{DC} , $n = -0.6 \times 10^{11} \text{ cm}^{-2}$, $B_{\perp} = 14 \text{ T}$ and $D = 0.15 \text{ Vnm}^{-1}$. Data points that correspond to the noise floor of the measurement setup and the 50 Hz electrical grid were removed (see Methods and reply to comment 4). The noise bulges are fitted with polynomial functions represented by dashes lines. Note that these measurements were conducted one year after those presented in Fig. 1 in the manuscript. While the overall noise level has changed, the frequency dependence remains consistent. **(b)** Dependence of the washboard frequency f_0 extracted from **(a)**. A linear fit is shown in grey.

Regarding other insulating states that are not Wigner phases, we have not observed any noise bulges. The corresponding data is shown in our response to comment 1 (Fig. R1) and is also shown and discussed in the revised manuscript (Extended Data Fig. 2), further supporting the distinct nature of noise behavior in the Wigner phase.

4. The data in Fig. 5, which purportedly provides the strongest signature of the Wigner crystal phase is not very high quality. The bulge is not visible in the raw data and appears only after normalization. This is unlike the GaAs experiments, where the noise peak is well-defined and does not need a background correction. At the very least, error bars should be provided for Figs. 5d and 5e, considering how broad these features are.

Reply:

We agree with the referee that the frequency dependent data is not of very high quality. In fact, the data quality is affected by additional sharp peaks, approximately 50 Hz and its higher harmonics, corresponding to electrical power distribution noise (the standard mains frequency in Germany is 50 Hz). This can be best seen when S is plotted as a function of frequency as follows:

Fig. R4. S as a function of f at $n = -1.35 \times 10^{11} \text{ cm}^{-2}$, $D = -0.7 \text{ Vnm}^{-1}$, $B = 0 \text{ T}$ and $I = 80 \text{ nA}$, plotted on a linear scale in f . Peaks at 50 Hz and its higher harmonics, corresponding to the 50 Hz electrical grid, are highlighted by red arrows.

To improve the presentation, we have removed data points that correspond to the noise floor of the measurement setup and the 50 Hz electrical grid in the revised manuscript. This has significantly improved the data presentation and the fitting of the frequencies. Exemplarily, we compare the original and revised Fig. 5 below:

Fig. R5. $S / S (I_{DC} = 80 \text{ nA})$ measured as a function of f at different I_{DC} before **(a)** and after **(b)** removing data points that correspond to the noise floor of the measurement setup and the 50 Hz electrical grid.

Furthermore, we have included a discussion of the data analysis procedure in the Methods section and included error bars in the revised version of the manuscript for Figs. 5d, 5e and Extended Data Fig. 1b.

5. Can the authors convincingly rule out contact effects? Insulating states in graphene are particularly prone to various transport artifacts because of contact equilibration issues. It would be important to show that the noise signatures don't arise from the contacts.

Reply:

We thank the referee for raising this important question regarding potential contact effects. We agree that poorly functioning contacts can lead to noise artifacts, and we have taken measures to exclude such effects. Specifically, we have investigated regions in the phase diagram with similar resistance and found no evidence of noise in those regions. For example, close to the band edge (green and black crosses in Fig. R6a) the conductance exhibits a similar value to that of phase II (the Wigner phase). However, conductance fluctuations are only observed within phase II (Fig. R6b, orange line) but not near the band edge (Fig. R6b, green and black lines).

Fig. R6. Observation of noise in AB-stacked bilayer graphene at zero magnetic field. **(a)** Conductance G as a function of charge carrier density n and out-of-plane magnetic field B_{\perp} at an electric displacement field at $D = -0.7 \text{ Vnm}^{-1}$ and an AC current of 78 Hz and 1 nA ($I_{DC} = 0 \text{ nA}$). The various phases are schematically illustrated and labeled in the mirror image according to A. M. Seiler et al., *Nature* **608**, 298-302 (2022). **(b)** Line cuts of the conductance as a function of time t at zero magnetic field for the selected densities indicated by the colored crosses in **(a)**: $n = -1.7 \times 10^{11} \text{ cm}^{-2}$ (purple, phase III), $n = -1.3 \times 10^{11} \text{ cm}^{-2}$ (orange, phase II), $n = -0.65 \times 10^{11} \text{ cm}^{-2}$ (green, quarter metal), and $n = -0.6 \times 10^{11} \text{ cm}^{-2}$ (black, quarter metal).

Additionally, as noted in our response to comment 1, no noise is observed within the quantum Hall states, further supporting that the observed noise signatures are intrinsic to phase II and not related to any contact equilibration issues.

We have now added the discussion on the contact effects to the revised manuscript.

In light of these issues, I cannot recommend publication in the present form of the manuscript.

Reply:

We thank the referee once again for the valuable feedback. By addressing the referee's questions and comments, we have significantly improved the quality and clarity of our manuscript. We hope that the revisions, particularly the improved data presentation and the additional experimental measurements, have addressed the referee's concerns. Furthermore, we have expanded our discussion to consider the possibility of Wigner solid. With these improvements, we hope the referee can now recommend our work.

Reviewer #3 (Remarks to the Author):

Seiler et al. report on studies of sliding Wigner crystals (WC) in the bilayer graphene system at zero and high magnetic fields. They primarily do so by revealing non-linear IV characteristics and noise in the posited WC regime of AB-stacked bilayer graphene.

Reply:

We thank the referee for summarizing our work and for providing detailed feedback. We are grateful for all the suggestions and comments, and we have addressed them below and incorporated the changes into the revised manuscript. All changes made in the manuscript are highlighted in blue below.

Comments:

The authors first show data in the high field regime, but I find the discussion of the data, which is restricted to one paragraph, inadequate. For example, the authors cite Ref. [5] as direct evidence for the presence of WC. In Ref. [5], the presence of WC is reported in a filling range $0.13 < \nu < 0.38$. However, the authors here find non-linear IV signatures only at $\nu < 0.2$. Why are the non-linear IV signatures not seen at higher filling factors? Further, the noise generated due to sliding vanishes by 2 nA. Why does this happen in the high field regime? The authors report the same behavior in the zero field case, and attribute it to phase slippage due to a large current. They should elaborate on this. Why is there a discrepancy in the maximum current needed for this phase slip at high (2 nA) and the zero field (4-5 nA) case?

Reply:

We thank the referee for the detailed comparison of our noise measurements with the results of Y.-C. Tsui et al. (ref. 5). Indeed, we observe non-linear IV characteristics only for $\nu < 0.2$, whereas ref. 5 reported WC signatures in the range $0.13 < \nu < 0.38$. However, in ref. 5 the WC was not observed at all intermediate filling factors—for instance, near $\nu = 0.33$, a competing fractional quantum Hall state (FQHS) emerges, which is also our case. Further differences between the two experiments may arise from differences in sample characteristics, such as disorder or screening effects (e.g., the sample in ref. 5 is not fully encapsulated), which could influence the stability and detection of the WC and other competing phases. For example, FQHS with lower filling fractions have been observed by other groups (e.g., $\nu = -2/7$ by J.I.A. Li et al. (Science 358, 648-652, (2017))) and would compete with the WC, but this is not the case for ref. 5.

Regarding the seemingly discrepancy of the noise bulges between the high-magnetic-field phase (4–5 nA) and the zero-magnetic-field phase (2 nA), we note that they are two distinct phases, though both exhibit WC features. Additionally, the strong magnetic

field may have some impacts on domain formation and scattering dynamics, which may reduce the WC's stability against the applied current, thereby lowering the threshold current in the process of depinning or phase slip.

To address the referee's suggestion, we have elaborated more on our observations in the high-field regime in the revised manuscript. We have also conducted additional measurements within the fractional quantum Hall state at $\nu = -1/3$, where we do not observe an increased noise level with increasing applied current. This stands in contrast to the Wigner crystal regime at lower densities, where a pronounced noise enhancement is seen and the peak frequencies of the noise bulges increase linearly with the applied current. These new measurements are now included in Extended Data Fig. 1 and Fig. 2 of the revised manuscript.

The conductances are in arbitrary units. The authors should show the exact values in the figures [for example Fig. 3(a)].

Reply:

We thank the referee for this comment. In our original submission, the conductance was presented in arbitrary units due to the finite contact resistance inherent in our two-terminal devices. In response to the referee's suggestion, we have now updated all figures to display the conductance \$G\$ in SI units.

Also, the authors should estimate the related threshold electric fields based on the conductance and the threshold current, and then estimate the domain size [based, e.g., on Phys. Rev. B 17, 535 (1978)]

Reply:

We thank the referee for this valuable suggestion. Based on PRL 131.236501 (2023) and PRB 17, 535 (1978), we have estimated the domain size in the presence of disorder. Using $L_0 = \text{sqrt}\left(\frac{0.02e}{4\pi\epsilon\epsilon_0 E_{th}}\right)$, we find $L_0 = 2 \mu\text{m}$, where the threshold electric field is $E_{th} = \frac{I_{th}}{l * G} = 2.5 \text{ V/m}$, with threshold current $I_{th} = 1 \text{ nA}$, sample length $l = 10 \mu\text{m}$ and conductance $G = 4 \times 10^{-5} \text{ 1}/\Omega$. We have included this estimation in the revised manuscript to discuss the discrepancy between the estimated and the experimentally determined frequencies. This issue is also discussed in more detail below in our response to the referee's fifth comment.

In Fig. 3(b), the conductance falls even further (about 25%) past the conductance fluctuation regime. If this is past the phase slip regime due to large currents, it is unclear why the conductance has to drop even lower than the pinned phase at very low I_{DC} .

Reply:

We thank the referee for this observation. The microscopic reason for the additional decrease in conductance past the conductance fluctuation regime remains unclear. One possible explanation is that the system enters a new phase as the WC is destroyed. Another possibility is that contact effects become more pronounced at higher bias currents, potentially leading to increased scattering and a further reduction in the measured conductance. However, we do not have a definitive understanding of this behavior at present.

The discrepancies associated with the washboard frequencies (~ 100 Hz and ~ 10 MHz), if attributed to domain sizes, would correspond to domains of the order $L = \llbracket 10^5 a \rrbracket_0$. ($f \sim v/L$ instead of v/a_0) which far exceeds the sample dimensions ($5 \mu\text{m}$), implying the presence of a single domain within the sample. But is this really the case? The authors should discuss this.

Reply:

We thank the referee for this insightful comment. We agree that, if the domain-size picture was responsible for our observation, the corresponding domain size would far exceed our sample dimension. In fact, we have estimated the domain size in the presence of disorder to be approximately $2 \mu\text{m}$ (see our reply to the third comment). It follows that the expected frequency for a sliding mode crossing a single domain ($f \sim v/L_0$ instead of v/a_0) is ~ 320 kHz, still much higher than the observed 100 Hz.

Therefore, the domain-size picture alone cannot account for the observed noise frequencies. Other possible explanations for the frequencies include variations in the current distribution and thus in the in-plane electric field (due to contact effects), thermal fluctuations, and overestimation of the average velocity. To address this referee's comment in the revised manuscript, we have expanded our discussion of the observed discrepancy, incorporating both our estimation of the domain size in the presence of disorder and other speculated explanations of the discrepancy.

Also, from Fig. 5(b,c), it can be gathered that the integrated noise in the very low-frequency regime continuously decreases. Can the authors comment on this, since I would expect the sliding noise to be nearly constant.

Reply:

We thank the referee for this observation. We have studied the very low-frequency regime in more detail and found that in the very low-frequency regime, the noise decreases with increasing I_{DC} when the frequency is small, whereas for higher frequencies the noise level stays nearly constant. To exemplify this, we here shown the noise level at $f = 13.7$ Hz as a function of the dc current at $B = 0$, $D = -0.7 \text{ Vnm}^{-1}$, and $n = -1.35 \times 10^{11} \text{ cm}^{-2}$ below.

Fig. R7. Noise level at $f = 13.7$ Hz as a function of the applied dc current at zero magnetic field B , an electric displacement field $D = -0.7 \text{ Vnm}^{-1}$ and a charge carrier density of $n = -1.35 \times 10^{11} \text{ cm}^{-2}$. The noise level stays constant for $I_{DC} \geq 3$ nA.

We note that the decrease in low-frequency noise with increasing I_{DC} was also found in other CDW systems (e.g., APL 118, 223101 (2021)) and a magnetic-field induced WC (PRL 131, 236501 (2023)). This general observation deserves future investigation.

Also, the differing slopes in Fig. 5(d) and 5(e) suggest that the domain sizes are different at different parallel fields. The authors should elaborate on why the domain size depends on parallel field.

Reply:

We thank the referee for pointing out this interesting difference and asking for the reason. In a previous work we demonstrated that the Wigner phase is in-plane spin polarized by observing the dominant in-plane magnetic hysteresis and the phase expansion under the in-plane magnetic field (see Nature 608, 298–302 (2022) for more details). All suggests that applying an in-plane magnetic field stabilizes the Wigner crystal, leading to the onset of sliding at a higher current level and the enhancement of the domain sizes, resulting in the observed different slopes in Fig. 5d and e. We have included this discussion into the revised manuscript.

General comment: I find, in general, the figure quality poor, especially the ones which show the conductance fluctuations. These must be improved.

Reply:

We agree with the referee that the frequency dependent data is not of very high quality. In fact, the data quality is affected by additional sharp peaks, approximately 50 Hz and its higher harmonics, corresponding to electrical power distribution noise (the standard mains frequency in Germany is 50 Hz). This can be best seen when S is plotted as a function of frequency as follows:

Fig. R4. S as a function of f at $n = -1.35 \times 10^{11} \text{ cm}^{-2}$, $D = -0.7 \text{ Vnm}^{-1}$, $B = 0 \text{ T}$ and $I = 80 \text{ nA}$, plotted on a linear scale in f . Peaks at 50 Hz and its higher harmonics, corresponding to the 50 Hz electrical grid, are highlighted by red arrows.

To improve the presentation, we have removed data points that correspond to the noise floor of the measurement setup and the 50 Hz electrical grid in the revised manuscript. This has significantly improved the data presentation and the fitting of the frequencies. Exemplarily, we compare the original and revised Fig. 5 below:

Fig. R5. $S / S (I_{DC} = 80 \text{ nA})$ measured as a function of f at different I_{DC} before **(a)** and after **(b)** removing data points that correspond to the noise floor of the measurement setup and the 50 Hz electrical grid.

Furthermore, we have included a discussion of the data analysis procedure in the Methods section and included error bars in the revised version of the manuscript for Figs. 5d, 5e and Extended Data Fig. 1b.

Summary: Although the authors show some potentially interesting data, I am not convinced that the observations provide clear evidence for sliding WC states. Hence, I cannot recommend publication.

Reply:

We thank the referee once again for the valuable input. In response to the concerns raised, we have undertaken substantial revisions to improve our work. Notably, we have expanded the discussion on the high magnetic field regime, addressed discrepancies associated with the washboard frequencies, and refined the presentation of the frequency-dependent data. We believe these comprehensive enhancements effectively address the referee's concerns and elevate the quality of our work, making it suitable for publication now.

Reviewer #4 (Remarks to the Author):

Reply:

We thank Reviewer #4 for co-reviewing our manuscript. The feedbacks provided by all the referees are highly constructive and have helped us improve the quality of our manuscript. We hope that our revisions address all the concerns and that the improved manuscript meets the expectations of all the reviewers.